# Vesselpose: Vessel Graph Reconstruction from Learned Voxel-wise Direction Vectors in 3D Vascular Images

**Rajalakshmi Palaniappan**[1,2]                    Rajalakshmi.Palaniappan@mdc-berlin.de
**Christoph Karg**[1]
**Nemesio Navarro-Arambula**[3]
**Peter Hirsch**[1,4] iD
**Dagmar Kainmueller**[*1,3,4] iD
**Lisa Mais**[*1,3,4] iD                         Lisa.Mais@mdc-berlin.de
[1] *Max-Delbrueck-Center for Molecular Medicine in the Helmholtz Association (MDC),*
[2] *Humboldt University of Berlin,* [3] *University of Potsdam,* [4] *Helmholtz Imaging*

**Editors:** Accepted for publication at MIDL 2026

## Abstract

Blood vessel segmentation and -tracing are essential tasks in many medical imaging applications. Although numerous methods exist, the prevailing segment-then-fix paradigm is fundamentally limited regarding its suitability for modelling the task of complete and topologically accurate vascular network reconstruction. We here propose an approach to extract topologically more accurate vascular graphs from 3D image data, building upon highly successful ideas from the related biomedical tasks of cell segmentation and -tracking. Our approach first predicts voxel-wise vessel direction vectors joint with standard vessel segmentation masks. Second, to extract the vascular graph from these predictions, we introduce a direction-vector-guided extension of the TEASAR algorithm. Our approach achieves state-of-the-art performance on three benchmark datasets, spanning both synthetic and real imagery. We further demonstrate the applicability of our approach to challenging 3D micro-CT scans of rat heart vasculature. Finally, we propose meaningful and interpretable measures of topological error, namely false splits and false merges for graphs. Overall, our approach substantially improves the topological accuracy of reconstructed vascular graphs, being able to separate closely apposed vessel segments and handle multiple vascular trees within a single volume.

**Keywords:** Blood Vessel Reconstruction, Centerline Topology, Evaluation.

## 1. Introduction

Tree-like structures are ubiquitous in living organisms and serve vital functions, for example as blood vessels, airways, or neuronal networks. Understanding how these branched systems develop, function, and change under pathological conditions requires detailed structural information. Image-based analysis has become a key tool for investigating such processes at the subcellular to organ scale, using diverse modalities such as MRI, micro-CT, electron microscopy, or light-sheet microscopy (Cheng et al., 2024; Walek et al., 2023; Damon-Soubeyrand et al., 2023; Todorov et al., 2020; Obenaus et al., 2017). To analyze these structures, image data are processed to extract simplified network representations—typically skeletons or graphs—from which features such as overall topology, segment

---

* Contributed equally

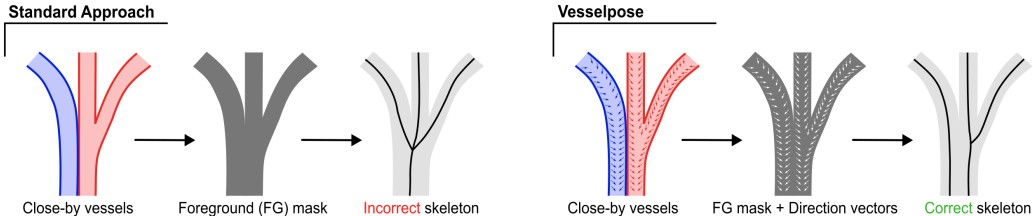

Figure 1: **Segmentation-and-skeletonization vs. Vesselpose.** Traditional segment-and-skeletonize pipelines often produce incorrect skeletons, especially when distinct vessels lie in close proximity. In contrast, Vesselpose leverages pixel-wise direction vectors to robustly reconstruct vascular trees, naturally handling closely apposed branches as well as multiple distinct trees.

lengths, and branching patterns can be quantified. For example, Liu et al. (2021) showed that altered topology of the microvasculature plays an important role in hepatocellular carcinoma (HCC). However, obtaining accurate and complete reconstructions remains challenging; manual or semi-automated tracing is still often the method of choice, particularly in cardiovascular imaging (Pampols-Perez et al., 2025; Rios Coronado et al., 2025).

Topological correctness refers to accurately preserving the connectivity of the biological network. Many vessel-graph extraction pipelines perform foreground-background segmentation of the image first (Tetteh et al., 2019; Todorov et al., 2020; Wittmann et al., 2025), and then skeletonize the foreground mask using TEASAR or variants of the Lee algorithm (Meyer-Spradow et al., 2009; Drees et al., 2021; Bumgarner and Nelson, 2022). However, these approaches struggle to achieve topological accuracy: Variable imaging contrast can render vessel segments faint or discontinuous, prompting segmentation methods to produce false splits; at the same time, branches running in close proximity often lead to false merges that incorrectly connect distinct vessels. In the subsequent skeletonization step, such false merge errors create artificial cycles or spurious branching points, as illustrated in the left part of Figure 1. Topological losses (Lux et al., 2025; Kirchhoff et al., 2024; Shit et al., 2021) or simple heuristics like thinning of ground-truth masks may help with this issue to some extent – However, segmentation as a modular step remains fundamentally ill-suited for modelling the task of topologically correct vessel graph reconstruction. This limitation is not specific to foreground-background segmentation, but also holds for instance segmentation: Although the vasculature forms a globally connected system, imaging typically covers only a restricted anatomical region, yielding multiple disjoint trees; while instance segmentation can, in principle, model the separation of different vessel trees, it is not designed to prevent false merges *within* a single tree.

Therefore, alternative strategies are needed to achieve reliable and topologically accurate vascular graph reconstructions. Earlier work used combinatorial optimization to assemble vessel trees from small centerline tracklets (Türetken et al., 2010, 2011), yielding globally optimal tree reconstructions w.r.t. some objective under topological constraints, thereby ensuring topological correctness. However, even with heuristics and relaxed constraints, Integer Linear Programming (ILP)-based methods (Türetken et al., 2016; Robben et al., 2014, 2016; Rempfler et al., 2016) remain computationally expensive and do not scale to large vascular networks. More recently, image-to-graph frameworks (Prabhakar et al., 2024;

Naeem et al., 2024, 2025), inspired by DETR (Carion et al., 2020; Zhu et al., 2021), have emerged as a promising direction. Most of them have been validated primarily on synthetic datasets, where Vesselformer (Prabhakar et al., 2024) still produces notable topological errors, while Trexplorer (Naeem et al., 2024) suffers from duplicate branching and premature tracking termination. Trexplorer-Super (Naeem et al., 2025) addresses these issues and extends evaluation to real datasets, yet its training and evaluation remain restricted to single-tree structures, whereas real vascular volumes typically contain multiple disjoint trees. Thus, a more general and computationally feasible solution is still required—one that can robustly extract topologically meaningful graphs from multi-tree vascular networks.

At the same time, issues of topological correctness have been addressed very successfully for the highly related tasks of cell segmentation and tracking in 3D(+t) microscopy data (Stringer et al., 2021; Malin-Mayor et al., 2023). Here, the community has moved away from the traditional segment-then-fix paradigm, replacing deep learning (DL)-based binary segmentation with models that predict pixel-wise shape properties that encode topologically relevant information (Hirsch and Kainmueller, 2020; Mais et al., 2020; Sheridan et al., 2023). Most prominently, Cellpose (Stringer et al., 2021; Pachitariu and Stringer, 2022) predicts vector fields pointing toward object centers; similarly, for cell tracking through time, Malin-Mayor et al. (2023) predict pixel-wise direction vectors that point backward in time to the center of the same or mother cell in the previous frame. Iteratively following these vectors reconstructs complete cell lineages and ultimately traces each cell back to its origin. This approach leverages the biological prior that cells divide but do not merge, ensuring a unique predecessor. These advances highlight the value of predicting pixel-wise topological information, suggesting a promising direction that has not yet been extended to vascular tree reconstruction.

Building on these insights, we propose a method that extracts topologically plausible vessel trees from 3D images using a heuristic solver guided by pixel-wise predictions. We train a network to predict direction vectors that point toward the vessel centerline while being biased in the rootward direction, leveraging the anatomical prior that vessel diameter typically increases toward the root. This prior enables robust orientation along the tree and naturally suits vascular and airway networks. By defining the flow from endpoints toward the root, we circumvent directional ambiguities at branching points, resulting in a well-defined direction vector at each location.

The predicted binary mask and direction vectors serve then as input to a novel skeletonization objective that reconstructs the tree structure by following the learned vector field. In summary, our main contributions are as follows:

- We present a DL-based method that predicts voxel-wise directional vectors from 3D vascular images, which a fast heuristic solver then assembles into a consistent vessel centerline graph.

- We introduce meaningful and easily interpretable topology-aware evaluation metrics such as false splits and false merges for graphs, proposing a tailor-made *assignment strategy* (cf. Maier-Hein et al. (2024)) based on hierarchical graph-matching.

- We outperform current state-of-the-art on synthetic and real datasets and extend evaluation to a widely used multi-tree dataset (Tetteh et al., 2019) and a real 3D micro-CT dataset, achieving superior topological accuracy and reconstruction quality.

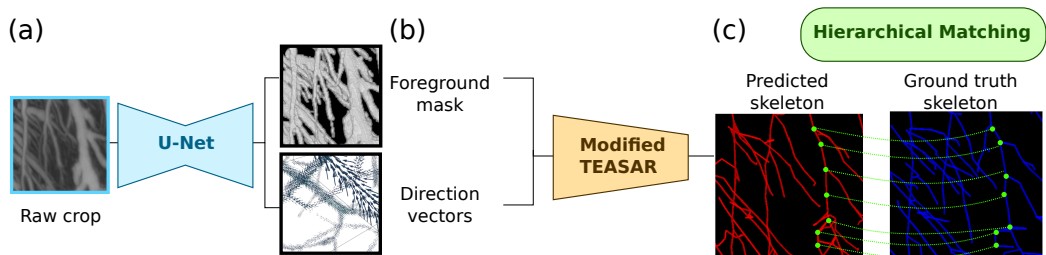

Figure 2: **Blood vessel reconstruction and evaluation.** (a) A U-Net predicts vessel foreground and voxel-wise direction vectors from the raw image. (b) A modified TEASAR algorithm extracts a skeleton graph. (c) Predicted skeletons are evaluated against ground-truth using hierarchical graph matching as assignment strategy, which yields topologically meaningful error metrics.

The code for the model and evaluation, along with trained models and prediction results, is publicly available at https://github.com/Kainmueller-Lab/Vesselpose.

## 2. Method

To derive graph representations from raw 3D vascular images, we first predict voxel-wise direction vectors capturing local structure, subsequently assembled into a tree-structured skeleton using a modified TEASAR algorithm (Sato et al., 2000). Figure 2 illustrates our approach. Section 2.1 describes direction vector generation and prediction, Section 2.2 the adaptation of TEASAR for vector-based skeletonization.

We formally represent vessel trees as directed acyclic graphs (DAG) (Diestel, 2017): Nodes correspond to 3D coordinates marking either branching points or sampled points along vessel segments, while edges denote the segments connecting them. Each edge is assigned a radius characterizing the local vessel thickness. Edge directions follow a parent–child relationship, where the parent is the one closer to the vessel root.

### 2.1. Direction Vector Generation and Prediction

We train a 3D U-Net (Ronneberger et al., 2015) to jointly predict a foreground mask and voxel-wise direction vectors (x, y, z components) from 3D grayscale images. These vectors point toward the vessel centerline and are additionally biased rootward by a fixed stepsize, such that iterative following of the vectors eventually converges at the root. Given a ground-truth foreground mask and a corresponding graph, we obtain training vectors as follows: For each foreground voxel, we first identify the nearest edge in the ground-truth graph within the local vessel radius. From the closest point on that edge, we then step a fixed distance toward the root, which defines the target point for the direction vector. The resulting vectors point from each voxel to this upstream centerline point, yielding smaller magnitudes for pixels close to the centerline and larger ones near the vessel boundary. Suppl. Figure 6a and Algorithm 1 provide details of the direction vectors and their generation. During training, we use Binary Cross-Entropy (BCE) loss for the foreground mask and Mean Squared Error (MSE) loss for the direction vectors.

## 2.2. TEASAR-based Centerline Generation

TEASAR (Sato et al., 2000) extracts tree-shaped skeletons from volumetric tubular segmentation masks by placing centerlines in regions that lie maximally far from the object boundary. It implements this by iteratively tracing shortest paths from a root to any voxel within the mask, applying a penalty that discourages paths from approaching the boundary. This penalty depends solely on boundary distance, thus it may produce incorrect skeletons when vessels run in parallel, as shown in Figure 1. To address this, we propose a modified TEASAR variant that incorporates predicted voxel-wise direction vectors. These vectors exhibit minimal magnitude and smallest angle relative to the centerline direction at voxels closest to the centerline. We therefore augment the penalty term with components based on both *vector magnitude* and *angular deviation*. This additional penalty helps disambiguate vessels that are spatially tangent but semantically distinct.

**Extended Penalty Term.** Formally, let $\Omega \subset \mathbb{R}^3$ be the 3D object (foreground mask) and $\partial\Omega$ be the object boundary (vessel surface). Let $p \in \Omega$ be the voxel inside the object located at the end of the shortest path $P$, constructed in a previous iteration step. By $N \subseteq \Omega$ we denote the set of all those adjacent voxels of $p$ are lying within the object. For each neighboring voxel $n \in N$, the *distance from boundary field (DBF)* of $n$, as leveraged by the original TEASAR, refers to the shortest Euclidean distance from voxel $n \in N$ to the nearest boundary point $b \in \partial\Omega$:

$$\text{DBF}(n) := \min_{b \in \partial\Omega} ||n - b|| \tag{1}$$

To incorporate directional guidance, we additionally consider the *vector magnitude field (VMF)*, defined at each voxel $n$ as the magnitude of its direction vector $v_n$: $\text{VMF}(n) := ||\boldsymbol{v}_n||$. Note that $\text{VMF}(n)$ is minimal if the voxel lies on the centerline. Furthermore we denote by $\theta(p, n) \in [0, 180]$ the angle (in degrees) between the direction vector $\boldsymbol{v}_p$ of $p$ and the relative direction vector $\boldsymbol{r} := n - p$ from $p$ to $n$ as shown in Suppl. Figure 6b. Again $\theta(p, n)$ is minimal if $n$ is located in direction of the predicted direction vector $\boldsymbol{v}_p$. The adapted penalty value we propose is given by:

$$\text{PV}_{\text{flow}}(p, n) = 1{,}000{,}000 \cdot \left( \left(1 - \frac{\text{DBF}(n)}{M_1}\right)^{16} + \left(\frac{\text{VMF}(n)}{M_2}\right)^{16} + \left(\frac{\theta(p, n)}{M_3}\right)^{16} \right) \tag{2}$$

where

$$M_1 = \max_{p \in \Omega}(\text{DBF}(p))^{1.01}, \ M_2 = \max_{p \in \Omega}(\text{VMF}(p))^{1.01}, \ M_3 = 180 \tag{3}$$

This directly adopts the original TEASAR penalty, augmenting it with VMF- and $\theta$-based terms of analogous form. Given this penalty, skeleton tracing proceeds as usual, starting from a most root-distant end point determined analogously as in original TEASAR, and appending the minimum-penalty neighboring node to the path until the root is reached.

**Multi-root Processing and Adaptive Masking.** Original TEASAR generates one skeleton per connected component of the binary mask. However, segmentation errors may merge distinct vessels into a single component, producing structures with multiple trees and therefore multiple roots. To address this, we extend TEASAR to support multiple

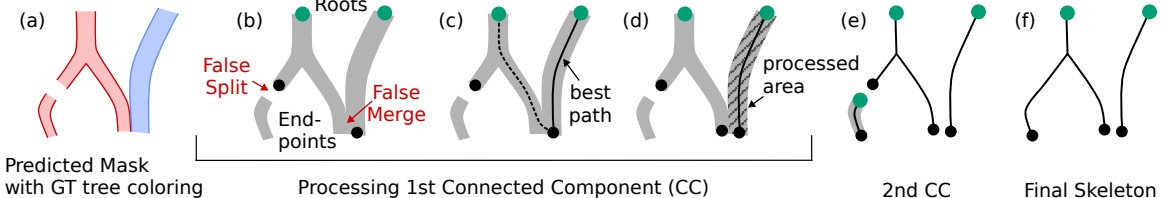

Figure 3: **Addressing topological errors with the modified TEASAR algorithm.** (a) Predicted foreground mask with distinct ground-truth (GT) trees having different colors. (b) The algorithm selects one connected component and identifies its roots and endpoints. (c) For each endpoint, paths are traced to all candidate roots, and the optimal path (with the lowest penalty) is chosen. (d) Voxels within a specified radius around the traced path are marked as processed and excluded from subsequent tracing. (e) After a component is fully processed, the algorithm proceeds to the next one; once all components are processed, disconnected fragments are evaluated for merging. (f) Final output with complete centerlines.

roots within a component, as illustrated in Figure 3. Root locations in the datasets are provided either by manual annotation or automatically using the predicted direction vectors, effectively speeding up the annotation process. Details on automated root detection are provided in Suppl. Section A.1.

Once the best path is established (Figure 3(c) and (d)), original TEASAR applies a simple linear thresholding using a fixed scale and constant value $d = scale \cdot r + const$, where $r$ is the vessel radius and $d$ is the masking distance used to exclude already processed regions. We extend this with an adaptive masking scheme in which both parameters vary smoothly with the local vessel radius. This makes the method more robust across vessels of different radius and effectively suppresses spurious small branches in larger vessels. Further details are provided in Suppl. Section A.2.

**False Split Postprocessing.** Finally, components without an assigned root are evaluated for potential merging with nearby trees. For each node in a disconnected component (the current node), we first identify neighboring nodes within a spatial distance of 5 voxels. Among these candidates, we retain only those that belong to a different tree and then compute their radius difference and angular deviation (based on the direction vectors) with respect to the current node. If the radius difference is below 3 and the angular difference is below 100 degrees, and adding an edge between the two nodes does not introduce a cycle, we connect them. In cases with multiple valid neighbors, we select the closest one. This step helps to reduce false splits introduced by the segmentation. However, very small or isolated components that do not meet these criteria may remain disconnected.

## 3. Evaluation

Our method predicts an acyclic vessel skeleton, with 3D coordinates assigned to each node, and edges oriented towards the root, forming a labeled directed acyclic graph (DAG). Comparing such graph against respective ground-truth (GT) is challenging (Drees et al., 2019;

Lyu et al., 2022): no standard metric exists (see Suppl. B.2), and many measures lack intuitive topological meaning or depend sensitively on node matching and sampling. To compare predicted and GT graphs, we first resample both at a fixed step size $s > 0$. The next essential step is an *assignment strategy* (cf. Maier-Hein et al. (2024)) that matches nodes and edges between predicted and GT graphs. In Sec. 3.1 we propose a greedy hierarchical matching procedure designed for robust topological correspondence. Based on these correspondences, we compute error metrics as described in Sec. 3.2.

### 3.1. Hierarchical Matching

Commonly used approaches to assign nodes or edges of two graphs to each other, is greedy nearest-neighbor- or optimal matching based solely on spatial proximity, such as in Drees et al. (2019); Naeem et al. (2025). While effective in simple scenarios, this strategy ignores the structural and semantic information inherent to tree-like graphs, making it unsuitable for capturing topological similarity—particularly in cases where different vessels are close-by. Therefore, we propose a greedy one-to-at-most-one hierarchical matching scheme that incorporates spatial, semantic and ancestor information. It is similar to Gillette et al. (2011), but also applicable in multi tree scenarios.

A pseudo-code description of the matching procedure is provided in Suppl. Algorithm 2. In short, first, the connected components of the GT graph $G$ and predicted graph $P$ are determined. Each node in $G$ and $P$ is assigned a semantic class—*root*, *branching point*, *leaf*, or *intermediate*. For every node in $G$, we identify candidate nearest neighbors in $P$ within a predefined distance threshold and rank them, first by semantic correspondence and second by spatial proximity. We then iterate over the GT roots, always selecting next the root whose best candidate exhibits the highest matching priority (i.e., first by identical semantic class and second by minimal distance). Starting from each root, we perform *two depth-first traversals*. In the first, we visit branching and leaf nodes, and assign them to the best available candidate based on the matching status of the candidate's parent, the candidate's semantic label, and its distance. Thus, candidates whose parents are matched within the same GT tree receive highest priority. If no suitable candidate exists, the GT node remains unmatched. The second traversal processes intermediate nodes using the same criteria. After completing a GT tree, we proceed to the next. Importantly, the candidate lists are updated immediately whenever two nodes become matched to maintain consistency throughout the hierarchy. A quantitative comparison with greedy nearest-neighbor and Hungarian matching is presented in Suppl. Table 4.

### 3.2. Metrics Definitions

Based on our literature review in Suppl. Section B.2, we report the *edge-wise* F1 score, as used in Drees et al. (2019, 2021), since the F1 score is a widely established and, in our view, easily interpretable measure of topological correctness when applied to edges. Yet F1 alone does not capture the structural impact of certain errors. For instance, a false positive edge connecting unrelated nodes can distort the topology far more than a shortcut to an ancestor (see Figure 4). To address this, we introduce *false splits* and *false merges* as additional topology-aware error measures, extending prior work (Matula et al., 2015; Mais et al., 2024) to multi-tree graphs where errors may arise both within and across trees.

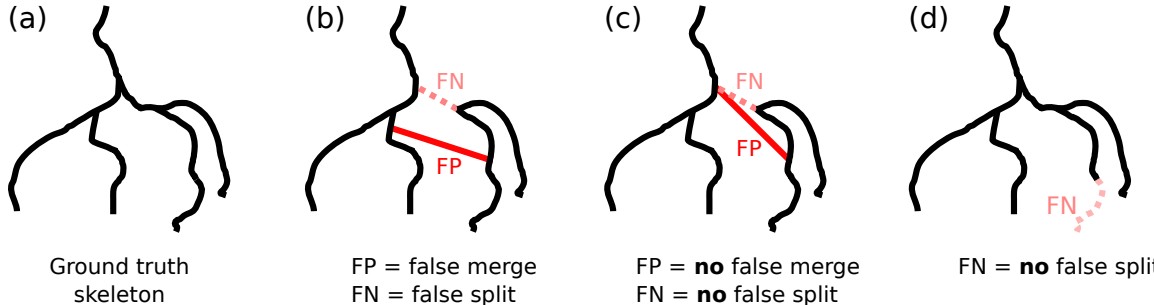

Figure 4: **False Merges & False Splits.** (a) A ground-truth skeleton next to three possible predictions. (b) The predicted graph has one FN and one FP edge. The FP is a false merge since it connects two nodes which are not ancestor of each other. Consequently, the FN is a false split. (c) The FP edge is *not* a false merge since it keeps the ancestor relation w.r.t. to its parent node intact. Consequently, the FN is *not* a false split. (d) The predicted graph only has FN edges which are *not* false splits since they do not change the node ancestor relations.

In addition, we report the metrics used in Naeem et al. (2025) for comparability reasons. Although they also report F1 scores at the node and branch level—similar in spirit to our recommendation—their metrics rely on greedy one-to-one nearest-neighbor matching and the computation operates on individual nodes, thereby not fully capturing connectivity. Regarding graph-level Betti numbers, only Betti–0 (the number of connected components) is meaningful; because assuming only trees, Betti–1 (the number of cycles) is always zero.

**Edge-wise F1 Score.** In the following, we denote by $G$ and $P$ the GT and predicted graphs with node sets $V_G$ and $V_P$, and by $\Phi : V_G \to V_P$ the one-to-one node matching. Both graphs are resampled to a fixed step size ($s = 1$ voxel, unless stated otherwise), after which we apply our proposed hierarchical matching. The edge-wise F1 score is computed as the balanced measure of precision and recall, relating the number of correctly matched edges (true positives, TP) in $G$ to the number of incorrectly matched (false positives, FP) or incorrectly unmatched (false negatives, FN) edges, where TP, FP and FN are defined as follows.

- An edge $(v, v')$ in $G$ is a TP if and only if $(\Phi(v), \Phi(v'))$ is an edge in $P$.

- An edge $(v, v')$ in $G$ is a FN if and only if $v$ and $v'$ were matched and $(\Phi(v), \Phi(v'))$ is *not* an edge in $P$.

- An edge $(\Phi(v), \Phi(v'))$ in $P$ is a FP if and only if $(v, v')$ is *not* an edge in $G$.

The *edge-wise F1, precision and recall* are defined as

$$F_1^{\text{edge}} := \frac{2\text{TP}}{2\text{TP} + \text{FP} + \text{FN}}, \quad Precision^{\text{edge}} := \frac{TP}{TP + FP}, \quad Recall^{\text{edge}} := \frac{TP}{TP + FN}$$

**False Merges (FM) and False Splits (FS).** We further define false merges and false splits as follows. A false merge is a FP edge $(\Phi(v), \Phi(v'))$ in $P$ where the GT nodes $v$ and $v'$ in $G$ have no directed path between them. In other words, neither is an ancestor of the other. For a false split, we consider the subgraph $P^*$ of $P$ obtained by excluding all false merge edges. A false split is a FN edge $(v, v')$ in $G$ such that adding the missing edge $(\Phi(v), \Phi(v'))$ to $P^*$ merges two connected components into one. This means FS correspond to missing edges in $P$ that cause false disconnections in $P^*$. The number of FS can be determined by $\beta_0(P^*) - \beta_0(G)$, since each FS increases the number of connected components of $P^*$. Here, $\beta_0$ is the number of connected components (Betti-0).

**Trexplorer-super Evaluation for Comparability.** To compute the metrics from Naeem et al. (2025), both $G$ and $P$ are resampled to one voxel spacing. At the node level, precision, recall, and F1 are reported, along with radius accuracy measured via the mean absolute error (MAE). At the branch level, the F1 score is reported; where a branch is considered a TP if at least 80% of its nodes are matched.

## 4. Experiments

We evaluate our method on four vascular datasets, covering both single and multi-tree scenarios. Consistent with prior works (Naeem et al., 2025, 2024; Prabhakar et al., 2024), all reported experiments use ground-truth root locations as input to our adapted TEASAR algorithm. Dataset-specific modifications of the training procedure, together with details on sample sizes and the data splits, are in Suppl. Section C. An ablation study quantifying the contributions of each components of our method (Suppl. Section C.5), a vector noise sensitivity study (Suppl. Section C.7) and test time noise sensitivity (Suppl. Section C.8) are also included. We also evaluate the effect of replacing the U-Net in our method with nnU-Net (Isensee et al., 2021) in Suppl. Section C.9.

### 4.1. Model Architecture and Training

For segmentation and vector prediction, we employ a 4-layer U-Net (Ronneberger et al., 2015) with batch normalization and 16 initial feature channels, which double at each down-sampling step. The network is trained on randomly sampled input patches. Data augmentation includes intensity shifts and randomly masking out $3 \times 3 \times 3$ pixel crops. Training is performed for 300,000 iterations with a batch size of 1 using the Adam optimizer. Aside from dataset-specific input sizes and augmentations, the architecture and training protocol are kept identical across all datasets. We have mentioned the further training details and different settings in Suppl. Section C.6.

### 4.2. Case 1: Single-Tree Data

For the single-tree datasets, we compare our method against Vesselformer (Prabhakar et al., 2024), Trexplorer (Naeem et al., 2024), and Trexplorer-super (Naeem et al., 2025). We report all baselines and metrics as in (Naeem et al., 2025), as we were unable to reproduce their published results and thus cannot faithfully compare in terms of our new metrics. Instead, we follow their evaluation protocol to enable a fair comparison. Thus we report point-level F1, precision, recall, and radius MAE, as well as branch-level F1 and Betti

Table 1: Quantitative comparison of our method with Vesselformer, Trexplorer and Trexplorer Super for the Single-Tree Synthetic and Parse 2022 datasets. Please note that we report all baselines and metrics as in Naeem et al. (2025). Our results are reported as mean and standard deviation ($\pm$) over three independent runs, while baseline results are reported over five runs.

| | Model | Point Level | | | | Branch Level | Graph Level | |
|---|---|---|---|---|---|---|---|---|
| | | F1↑ | Prec↑ | Rec↑ | Rad.(MAE)↓ | F1↑ | $\beta_0 \downarrow$ | $\beta_1 \downarrow$ |
| Synthetic | Vesselformer | $48.18_{\pm 5.62}$ | $44.53_{\pm 7.87}$ | $61.52_{\pm 1.14}$ | $0.42_{\pm 0.01}$ | $15.95_{\pm 0.36}$ | $81.7_{\pm 16.8}$ | $653.5_{\pm 138.7}$ |
| | Trexplorer | $39.40_{\pm 8.62}$ | $30.91_{\pm 9.45}$ | $78.21_{\pm 4.13}$ | $0.23_{\pm 0.03}$ | $26.26_{\pm 7.18}$ | $0_{\pm 0.0}$ | $0_{\pm 0.0}$ |
| | Trexpl. Super | $77.83_{\pm 1.89}$ | $91.91_{\pm 3.28}$ | $70.44_{\pm 3.02}$ | $\mathbf{0.1}_{\pm 0.01}$ | $77.12_{\pm 1.59}$ | $0_{\pm 0.0}$ | $0_{\pm 0.0}$ |
| | **Ours** | $\mathbf{92.25}_{\pm 0.02}$ | $\mathbf{95.49}_{\pm 0.01}$ | $\mathbf{89.24}_{\pm 0.05}$ | $0.29_{\pm 0.00}$ | $\mathbf{81.50}_{\pm 0.16}$ | $0_{\pm 0.0}$ | $0_{\pm 0.0}$ |
| Parse2022 | Vesselformer | $16.43_{\pm 0.78}$ | $18.49_{\pm 1.84}$ | $15.28_{\pm 0.83}$ | $1.11_{\pm 0.03}$ | $1.99_{\pm 0.16}$ | $410_{\pm 23.9}$ | $246.7_{\pm 78.1}$ |
| | Trexplorer | $10.01_{\pm 4.98}$ | $9.87_{\pm 3.76}$ | $12.01_{\pm 7.46}$ | $1.21_{\pm 0.30}$ | $3.71_{\pm 1.91}$ | $0_{\pm 0.0}$ | $0_{\pm 0.0}$ |
| | Trexpl. Super | $39.46_{\pm 1.93}$ | $55.27_{\pm 3.00}$ | $33.99_{\pm 3.34}$ | $\mathbf{0.56}_{\pm 0.01}$ | $23.46_{\pm 1.09}$ | $0_{\pm 0.0}$ | $0_{\pm 0.0}$ |
| | **Ours** | $\mathbf{57.52}_{\pm 0.66}$ | $\mathbf{59.11}_{\pm 0.37}$ | $\mathbf{57.81}_{\pm 0.89}$ | $0.58_{\pm 0.02}$ | $\mathbf{35.33}_{\pm 1.19}$ | $1.85_{\pm 0.46}$ | $0_{\pm 0.0}$ |

Table 2: Quantitative results of our method on the Single-Tree datasets using our proposed evaluation metrics to support future benchmarking. We report mean and standard deviation ($\pm$) over three independent runs.

| Dataset | Edges | | | FM↓ | | FS↓ | |
|---|---|---|---|---|---|---|---|
| | F1↑ | Prec↑ | Rec↑ | Rel. | Abs. | Rel. | Abs. |
| Synthetic | $0.89_{\pm 0.001}$ | $0.93_{\pm 0.001}$ | $0.87_{\pm 0.001}$ | $0.010_{\pm 0.0}$ | $25.86_{\pm 0.10}$ | $0.009_{\pm 0.0}$ | $25.86_{\pm 0.10}$ |
| Parse2022 | $0.69_{\pm 0.015}$ | $0.90_{\pm 0.004}$ | $0.57_{\pm 0.012}$ | $0.007_{\pm 0.0}$ | $83.8_{\pm 1.37}$ | $0.004_{\pm 0.0}$ | $85.62_{\pm 1.68}$ |

scores in Table 1. Apart from that, we also evaluated the single-tree datasets using our own metrics in Table 2 to support future benchmarking.

The **Single-Tree Synthetic** dataset, introduced in Naeem et al. (2025), is generated using the Synthetic Vascular Toolkit (SVT) (Sexton et al., 2025). Each volume contains a single vascular tree, its segmentation mask, and the corresponding 3D centerline graph. As shown in Table 1, our model consistently outperforms the current state-of-the-art across both point-level and branch-level metrics. Suppl. Figure 7 shows qualitative results.

The publicly available **Parse 2022** pulmonary artery segmentation dataset (Luo et al., 2024) contains 100 computed tomography pulmonary angiography (CTPA) volumes with pixel-wise segmentation masks. These masks were created semi-automatically by experts using a region-growing approach. Naeem et al. (2025) subsequently derived centerlines from these masks using the Kimimaro TEASAR implementation (Silversmith et al., 2021). Note that these ground-truth centerlines were generated *automatically*. As shown in Table 1, our model outperforms the current state-of-the-art on both point-level and branch-level F1. However, at graph level we note some Betti-0 errors: although our post-processing step is designed to correct false splits, it does not fully guarantee global connectivity of the predicted vascular tree. Suppl. Figure 8 shows qualitative results. We note that a potential bias may favor our method, since the ground-truth skeletons are generated using the TEASAR algorithm.

### 4.3. Case 2: Multi-Tree Data

Although vascular networks are ideally single-tree structures, real data often contain multiple trees due to challenges in separating arteries and veins or imaging artifacts. Here, we report our recommended metrics, namely edge-level F1, precision, recall, false merges (FM), and false splits (FS). We compare Vesselpose against different segmentation-based approaches, where we skeletonize the resulting binary masks with Kimimaro TEASAR.

The **Multi-Tree Synthetic** dataset originates from Tetteh et al. (2019) and is generated using vessel formation simulations (Schneider et al., 2012). We compare our method to vesselFM (Wittmann et al., 2025) and a standard U-Net (Ronneberger et al., 2015). We also report an upper bound by applying TEASAR directly to the ground-truth masks of Schneider et al. (2012). The results in Table 3 show that our method consistently outperforms these segmentation-based baselines. Original TEASAR produces one tree per connected component, but baseline segmentations frequently merge distinct trees into a single component. As a result, their false merge rates are substantially higher than our method. In Figure 5 we show qualitative results and discuss failure cases of our method.

The **Multi-Tree Micro-CT** data were acquired from perfused rat hearts using a solidifying Microfil contrast agent, using a protocol broadly similar to that described in (Napieczyńska et al., 2024). This approach provides strong vascular contrast and enables visualization of small vessels. The dataset is still under study and may be made publicly available at a later stage. We use four rat heart volumes: one to fine-tune a U-Net pretrained on the synthetic multi-tree data, and three for validation and testing. For these, we annotated three $400 \times 400 \times 400$ pixel crops using CATMAID (Saalfeld et al., 2009; Schneider-Mizell et al., 2016). Details on the data, model, and fine-tuning procedure are provided in Suppl. Section C.4. Table 3 shows that our method consistently outperforms a standard U-Net with TEASAR. Although the dataset is relatively small, the observed performance improvement is consistent with those reported on the other datasets. Our higher absolute FM and FS values stem from reconstructing more complete skeletons, whereas U-Net and regular TEASAR miss large graph regions—reflected in our correspondingly lower relative FM/FS counts, also seen in Suppl. Figure 9.

## 5. Conclusion

In this work, we presented a novel method for extracting vessel graphs from 3D images. We demonstrated its effectiveness on four datasets, comprising synthetic and real data, as well as single-tree and multi-tree vascular structures. In addition, we introduced a hierarchical graph matching algorithm that yields more topologically meaningful node and edge correspondences, and we defined false splits and false merges as intuitive topology-aware error measures for tree graphs. Despite these advances, several evaluation metrics such as the edge-wise F1 score remain sensitive to different node sampling and matching strategies. A more systematic analysis of these effects represents a valuable direction for future work. Moreover, there is a pressing need for publicly available, real-world datasets with high-quality, manually annotated 3D vessel graphs. Such datasets, combined with an established evaluation protocol encompassing sampling, matching, and metrics, would be essential for enabling consistent benchmarking and further methodological development in this area.

Table 3: Quantitative comparison of our method on the Multi-Tree Synthetic and Micro-CT Heart datasets. We compare against U-Net, vesselFM, and the ground-truth (GT) segmentation, each followed by TEASAR skeletonization. Because a more complete prediction can yield higher absolute FM and FS counts than an incomplete graph, we additionally report relative FM/FS values, obtained by dividing the absolute counts by the total number of predicted edges. We report mean and standard deviation ($\pm$) over three independent runs (except for vesselFM and GT).

| | Method | Edges | | | FM↓ | | FS↓ | |
|---|---|---|---|---|---|---|---|---|
| | | F1↑ | Prec↑ | Rec↑ | Rel. | Abs. | Rel. | Abs. |
| Multi-Tree Synthetic | U-Net | $0.46\pm0.001$ | $0.64\pm0.002$ | $0.36\pm0.001$ | $0.02\pm0$ | $51.87\pm0.68$ | $0.01\pm0.002$ | $51.70\pm3.39$ |
| | VesselFM | 0.46 | 0.62 | 0.36 | 0.02 | 51.3 | 0.01 | 58.3 |
| | GT Segm | 0.46 | 0.64 | 0.36 | 0.02 | 52.1 | 0.01 | 38.4 |
| | Ours | $\mathbf{0.80}\pm0.002$ | $\mathbf{0.79}\pm0.002$ | $\mathbf{0.80}\pm0.001$ | $\mathbf{0.007}\pm0$ | $\mathbf{30.80}\pm1.10$ | $\mathbf{0.007}\pm0$ | $\mathbf{29.67}\pm1.80$ |
| Micro CT | U-Net | $0.32\pm0.03$ | $0.22\pm0.04$ | $0.57\pm0.01$ | $0.01\pm0$ | $\mathbf{26.25}\pm2.25$ | $0.009\pm0$ | $\mathbf{23.5}\pm4.2$ |
| | Ours | $\mathbf{0.50}\pm0.002$ | $\mathbf{0.43}\pm0.001$ | $\mathbf{0.63}\pm0.002$ | $\mathbf{0.006}\pm0$ | $45.5\pm1.4$ | $\mathbf{0.006}\pm0$ | $42.5\pm1.4$ |

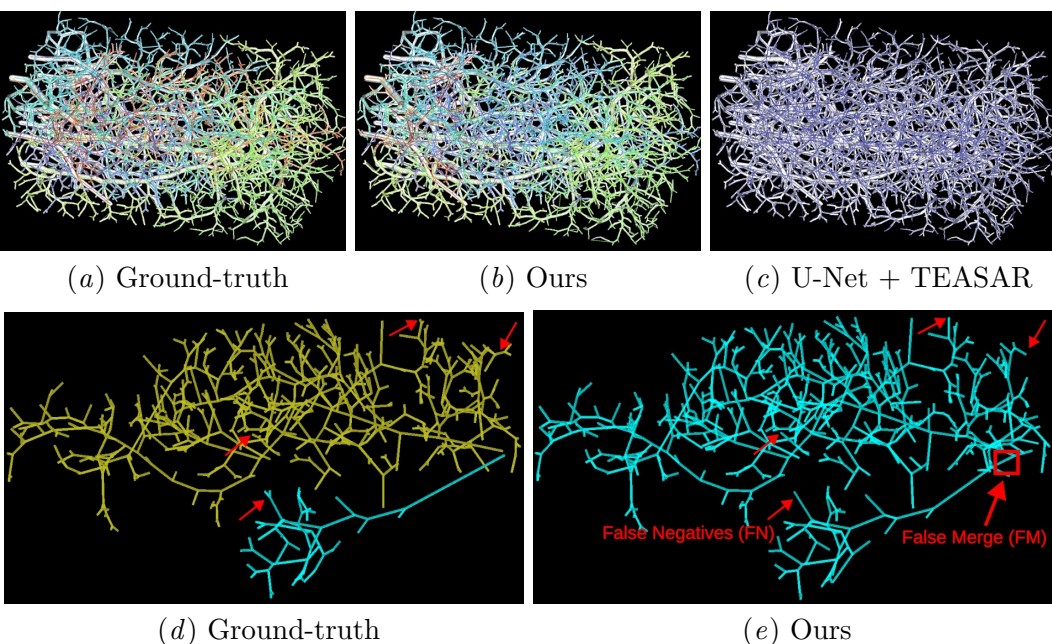

$(a)$ Ground-truth     $(b)$ Ours     $(c)$ U-Net + TEASAR

$(d)$ Ground-truth          $(e)$ Ours

Figure 5: **Qualitative results for the multi-tree synthetic dataset.** First row: Segmentation mask and skeletons overlaid, where each color represents a distinct tree. Our approach separates most trees, whereas U-Net + TEASAR merge all trees into one component. Second row: Failure cases for our method, including missed small terminal branches (red arrows) and falsely merged trees (red rectangle).

## Acknowledgments

This work was supported by the Berlin Center for Translational Vascular Biomedicine through the project "Post-preeclampsia: common mechanisms in microvascular dysfunction cause long-term risks for cardiovascular and retinal end-organ damage" and by the German Research Foundation (DFG) through the Individual Research Grant UMDISTO (project no. 498181230). We thank Kristin Kraeker, Hanna Napieczyńska and the Animal Phenotyping Platform of the Max Delbrück center for kindly providing unpublished rat heart data. We thank the Kainmueller Lab for their support and feedback.

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

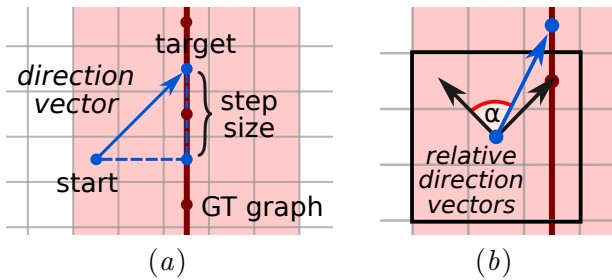

Figure 6: **Direction vectors generation and angular difference penalty used in modified TEASAR.** (a) Direction vectors (blue) are generated by first identifying the closest point on the ground-truth graph (dark red) and then stepping a fixed distance toward the root along the graph. Since the step size is constant across all voxel locations, direction vectors near the centerline exhibit smaller magnitudes, while those near the vessel boundary have larger magnitudes. (b) A penalty is assigned based on the angular difference between the predicted direction vector at the current voxel and the relative direction vector to each of its neighboring voxels (i.e., the "walk direction"). Lower penalties correspond to stronger alignment between the predicted and actual tracing directions, encouraging paths that follow the learned vessel orientation.

## Appendix A. Extended Methods

### A.1. Automated Root Detection

Vesselpose extracts a centerline graph using an adaptation of the TEASAR algorithm, which requires an initial set of vessel roots. In practice, these roots are annotated manually, a procedure that is both slow and labor-intensive. An automated strategy would remove this bottleneck, but detecting roots directly from the binary foreground mask is unreliable.

The voxel-wise direction field predicted by Vesselpose provides a direct workaround: roots correspond to sinks of this 3D flow field. A virtual particle is hereby initialized at every foreground voxel $p$ whose distance to the background is at least $r_{\min}$. The particle position is set to $x_0 := p$ and updated iteratively by $x_i := x_{i-1} + \lambda\, v_{i-1}$, where $v_{i-1}$ is the interpolated direction vector at $x_{i-1}$ and $\lambda > 0$ is a step-size parameter. A position $x_i$ is considered a sink when the displacement magnitude $\lambda\|v_i\|$ falls below a tolerance $\tau > 0$. After $N$ iterations, all detected sinks are collected and used as candidate vessel roots for initializing the adapted TEASAR procedure.

To evaluate the effectiveness of automated root detection we apply this method to the three validation data of the Multi-tree synthetic dataset and report how many manually annotated ground-truth roots are correctly detected. A root is considered to be correctly detected if it lies within a distance of 2 pixels from a calculated sink. For our experiments we further chose the following parameters: number of steps $N = 50$, step size $\lambda = 1.0$, tolerance $\tau = 0.1$ and min-radius threshold $r_{\min} = 3.0$.

As a result, the automated root detection algorithm is able to detect *all* of the ground-truth roots present in all three volume samples, while producing additional false positive

---

**Algorithm 1** Direction Vector Generation

---

**Input** : Vessel segmentation mask $M$ and the corresponding skeleton graph $G = (V, E)$ with each edge $e \in E$ having a radius $r_e > 0$ assigned

**Parameters:** `step_size` $> 0$

**Output:** `vector_field` of direction vectors

Fit $G$ to $M$ by rescaling

Find leaf nodes in $G$

Select the root of each connected component to be the leaf node with maximal radius

Direct the edges in each connected component tree such that they are directed towards the root.

**foreach** *foreground_voxel* of $M$ **do**
  | Find `nearest_edge` $\in E$ to `foreground_voxel` in $G$
  | Find `closest_point` to `foreground_voxel` on `nearest_edge`
  | $d \leftarrow$ distance between `foreground_voxel` and `closest_point`
  | **if** $d \leq r_{nearest\_edge}$ **then**
  | | `target_point` $\leftarrow$ move `closest_point` for `step_size` along $G$ towards the root
  | | `direction_vector` $\leftarrow$ `target_point` - `foreground_voxel`
  | | Store `direction_vector` in `vector_field`
  | **end**
**end**

**return** *vector_field*

---

roots. However, the additional false positive roots do not pose a problem to the TEASAR-based centerline generation since these can be filtered out during multi-root processing as described in 2.2.

## A.2. Adaptive masking

Standard TEASAR excludes processed areas using simple linear thresholding with a fixed scale and constant. However, in vesselpose these parameters vary with respect to the local radius as described in 2.2. Given predefined parameter ranges for $scale \in [s_{\min}, s_{\max}]$ and $const \in [c_{\min}, c_{\max}]$, and radius bounds $[r_{\min}, r_{\max}]$, we compute the normalized radius fraction

$$\alpha_r = \frac{r - r_{\min}}{r_{\max} - r_{\min}} \tag{4}$$

The scale and constant are then interpolated:

$$\text{scale}(r) = s_{\min} + \alpha_r(s_{\max} - s_{\min}) \ , \ \text{const}(r) = c_{\min} + \alpha_r(c_{\max} - c_{\min}) \tag{5}$$

The adaptive masking distance is finally defined as

$$d(r) = \text{scale}(r) \cdot r + \text{const}(r) \tag{6}$$

## Appendix B. Extended Evaluation

### B.1. Hierarchical Matching

In Algorithm 2 we demonstrate the steps involved in our hierarchical matching which takes the node label and its semantic into consideration rather than just the spatial proximity.

### B.2. Related work: Commonly used metrics

A unified approach for comparing graph structures is still lacking Lyu et al. (2022), and numerous evaluation strategies have emerged across the biomedical literature. On one hand, there are *detection- and segmentation-based metrics* for comparing centerlines, branching points, or graph edges, which we find unsuitable for graph-based evaluation: For example, clDice (Shit et al., 2021) does not reflect topology, as it operates at the pixel level and fails to penalize structural changes like loops or disconnections—errors that may drastically alter topology but only minimally impact clDice scores as few pixels are missing or added. The same limitation applies to precision, recall, and F1 scores when applied on pixel level. If mean average precision (mAP)(Lin et al., 2014) is used, as in Prabhakar et al. (2024); Naeem et al. (2024), nodes and edges are compared based on their overlap of the bounding boxes. However, as shown in Foucart et al. (2023), intersection-over-union (IoU) is not well-suited for small objects, especially in 3d, where meeting high IoU thresholds becomes impractical.

On the other hand, *graph similarity measures* are commonly used in related studies (Prabhakar et al., 2024; Naeem et al., 2024; Drees et al., 2019, 2021). However, the Street Mover's Distance (SMD) does not preserve connectivity information as it converts graph to point clouds for distance computation. Additionally, SMD is sensitive to resampling and hyperparameters, making it less reliable for topological evaluation. In contrast, Betti numbers provide a topological perspective, but, e.g., Betti-0 is too coarse, as it does not capture topological errors within individual trees (Lux et al., 2025). Instead, precision, recall, and F1 scores—when applied at the graph level, particularly for edges, as in Drees et al. (2019, 2021)—provide meaningful insights into topological correctness. However, these metrics do not account for the structural impact of individual errors: missing or spurious edges may vary greatly in severity depending on how drastically they alter the topology of the graph, a nuance not captured by the F1 score.

Another family of metrics includes *tree edit distance (TED)-based measures* like Li et al. (2023); Matula et al. (2015), which quantify the number of operations (e.g., node or edge insertions/deletions) needed to transform the predicted graph into the ground truth. These metrics offer an intuitive and meaningful notion of similarity, but we have not seen them commonly used in related work. For example, Li et al. (2023) introduces a TED variant specifically for vasculature graphs, but it penalizes heavily false merges and splits between two trees as it penalizes the falsely merged tree twice by adding the counts for removing every falsely added node and edge; and then for creating the missing tree from scratch. Another intuitive measure is the TRA metric (Matula et al., 2015), originally designed for evaluating cell lineages, but it is not directly applicable here as it relies on IoU-based matching.

---

**Algorithm 2** Hierarchical matching between $G$ and $P$

---

**Input**   : Directed acyclic graphs $G$ and $P$; Each node in each graph labelled as belonging to semantic class "root", "leaf", "branching point" or "intermediate point"; Each node in each graph has an associated 3d position; maximum distance for two points being matched $d_{max}$

**Output:** `match_dict`

Initialize empty `match_dict`

**foreach** *node $v$ in $G$* **do**
 Determine the set $W_v$ of closest nodes in $P$ that lie within distance $d_{max}$ to $v$
 Determine subset $W_{v,sem} \subseteq W_v$ of nodes with same semantic class as $v$
 Sort each subset by distance to $v$
 Append to form sorted list $W_v^{sorted} = [W_{v,sem},\ W_v \backslash W_{v,sem}]$
**end**

Sort root nodes $r$ of $G$ analogously, i.e.: Primary sorting criterion: non-empty $W_{r,sem}$ before all others; secondary criterion: distance of closest $w \in W_r$

**foreach** *$r$ in sorted list of root nodes in $G$* **do**
 **foreach** *class label $c$ in sorted list $["root","branching\ point"\ or"leaf","intermediate\ point"]$*
  **do**
   **foreach** *node $v$ with class label $c$ in depth-first traversal of $G$ from $r$* **do**
    **if** *class label of $v$ is "root"* **then**
     remove all elements $w$ in $W_v^{sorted}$ if parent of $w$ is already matched to another tree than $r$
    **end**
    **if** *$W_v^{sorted}$ is not empty* **then**
     **if** *class label of $v$ is no "root"* **then**
      get mask $m_v$ with true elements if parent of $w \in W_v^{sorted}$ is matched to same tree as $r$
      **if** *$W_v^{sorted}[m_v]$ is not empty* **then**
       update $W_v^{sorted}$ with $W_v^{sorted}[m_v]$
      **end**
     **end**
     pick first $w$ in $W_v^{sorted}$
     store $(v, w)$ in `match_dict`
     **foreach** *not yet visited node $v'$ in $G$* **do**
      remove $w$ from $W_{v'}^{sorted}$
     **end**
    **end**
   **end**
  **end**
 **end**
**end**
**return** `match_dict`

---

Our observations are consistent with Lyu et al. (2022), who conclude that selecting a reliable and unbiased evaluation metric remains an open problem in the community. With

Table 4: Quantitative comparison of our hierarchical matching with greedy one-to-one matching and optimal Hungarian matching. Lower false merge (FM) and false split (FS) values indicate better topology preservation during matching. Results are shown for Vesselpose on the validation set of the Multi-Tree Synthetic dataset. For this analysis, graphs were resampled to include only roots, branching points, and end nodes.

| Matching | Edges | | | FM | | FS | |
|---|---|---|---|---|---|---|---|
| | F1↑ | Prec↑ | Rec↑ | Rel.↓ | Abs.↓ | Rel.↓ | Abs.↓ |
| Greedy | 0.8 | 0.81 | 0.79 | 0.01 | 49 | 0.01 | 48.6 |
| Hungarian | **0.81** | **0.82** | **0.80** | 0.02 | 62.6 | 0.01 | 62.3 |
| Hierarchical (Ours) | 0.80 | 0.81 | 0.79 | **0.007** | **29.7** | **0.007** | **29.3** |

this work, we aim to contribute to this discussion by proposing a greedy hierarchical matching strategy and introducing false splits and false merges as topology-aware measures for robust assessment of vessel graph reconstructions. Nevertheless, we believe that a more systematic analysis of how node matching and sampling choices influence evaluation outcomes is still needed to establish common recommendations—an important direction for future work, but beyond the scope of this paper.

## Appendix C. Extended Experiments

We conducted all experiments on an HPC cluster using a single NVIDIA H100 GPU. Each model is trained for approximately 3 days. We allocate 200 GB RAM per job, although CPU and memory demands are modest since training relied mainly on GPU computation with Zarr-based random crop loading. The models are trained using PyTorch on a CUDA-enabled Linux environment.

### C.1. Single-Tree Synthetic Data

The dataset comprises 500 volumes of size $256^3$ voxels and is split into training, validation, and test sets. The training, validation and test split contain 368, 32 and 100 samples respectively. The network is trained on randomly sampled input patches of size $128^3$ voxels with intensity shift augmentation.

### C.2. Parse 2022 Challenge

The data has a in-plane size of $512 \times 512$ pixels and its z-stack comprises between 295 and 390 slices. The training, validation and test split contain 72, 8 and 20 samples respectively. Following Naeem et al. (2025), all volumes are resampled to an isotropic resolution of $0.5mm$. For both training and inference, we use input sizes of $256^3$ voxels and do not apply any data augmentation.

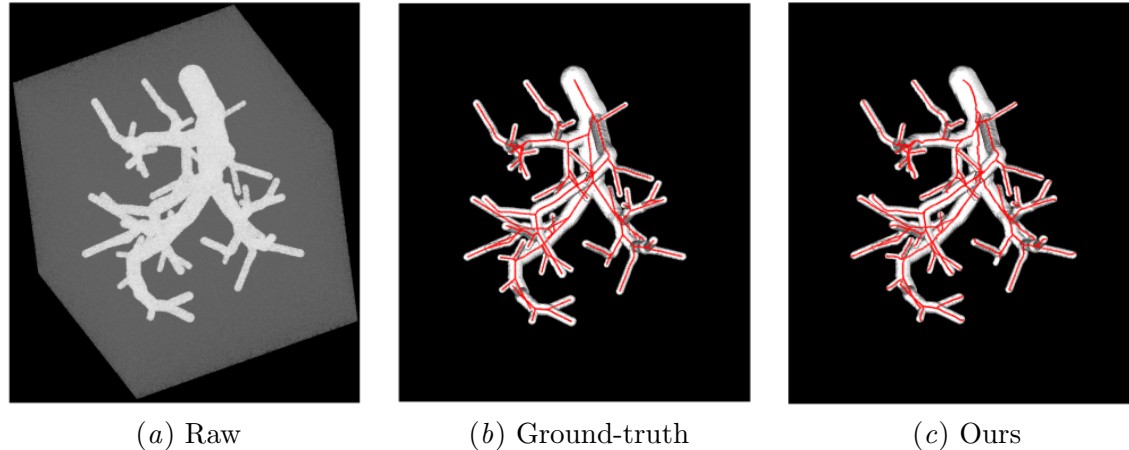

(a) Raw         (b) Ground-truth         (c) Ours

Figure 7: **Qualitative comparison for single-tree synthetic.** A 3D rendering of one of the samples. (a) shows the raw image (b) Ground-truth skeletons overlaid on the segmentation mask (c) our predicted skeleton overlaid on the predicted binary segmentation. Our method (c) produces a reconstruction that closely matches the ground-truth (b), capturing fine structures and maintaining topological consistency.

### C.3. Multi-tree Synthetic Data

This dataset is obtained from (Tetteh et al., 2019). Each volume measures $325 \times 304 \times 600$ voxels. We follow the information provided in Naeem et al. (2024); Prabhakar et al. (2024) about how many samples where used in which data split and utilized the first 50 volumes. The training, validation and test split contain 37, 3 and 10 samples respectively. Training and inference are performed with a input size of $256^3$ voxels, using intensity shifts and masked out crops as data augmentations. For this analysis, graphs are resampled to include only roots, branching points, and end nodes.

### C.4. Micro-CT Heart data

This dataset contains samples from both preeclamptic and healthy animals. The individual samples measure approximately $1300 \times 1300 \times 1700$ voxels at a resolution of $12\mu m$. To obtain foreground masks for fine-tuning the U-Net on the micro-CT heart data, we train a random forest classifier (Breiman, 2001) with 100 trees and a maximum depth of 10. We include Frangi vesselness features (Frangi et al., 2000) as additional input features for the random forest classifier and manually annotate a subset of vessels as ground truth. We then used the resulting foreground masks to fine-tune the U-Net model which was pre-trained with the synthetic multi-tree dataset. We fine-tune the U-Net by freezing all but the final layer to predict the foreground mask.

A key challenge associated with this dataset is its large size and the presence of substantial heart chambers, which occupy much of the volume. Thus, we downsample the data by a factor of 0.5 in each dimension and apply a heuristic approach to remove the chambers

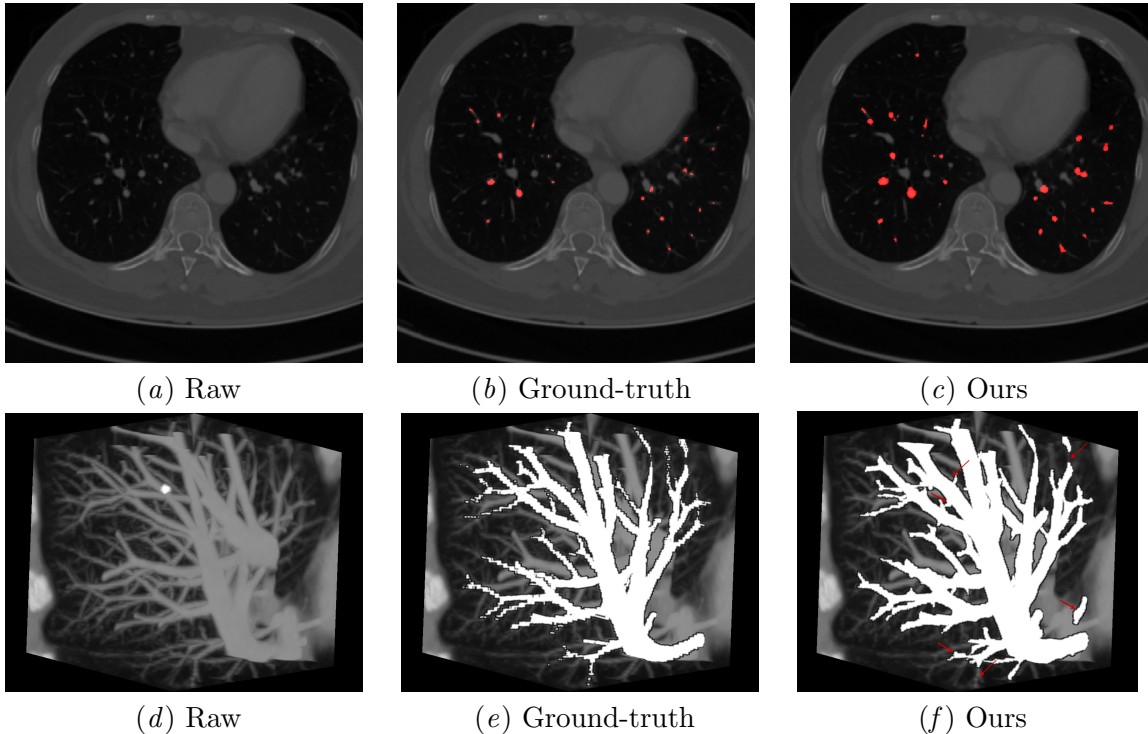

Figure 8: **Qualitative comparison of PARSE2022 segmentation** First row: A 2D slice from one sample. (a) shows the raw image, while (b) and (c) show the raw image overlaid with the provided ground-truth segmentation and our segmentation, respectively. Second row: A 3D crop from the same sample. (d) shows the raw image, and (e) and (f) show the raw image overlaid with the ground-truth and our segmentation, respectively. Both segmentation masks miss vessel segments that are visible in the raw data and contain disconnected components. Some of the failure cases in our segmentation is highlighted in red arrows(f).

by eliminating the largest connected component in each 2D segmentation slice. Fine-tuning and inference are performed with an input size of $256^3$ voxels.

## C.5. Ablation experiments in modified TEASAR

To assess the contribution of each modification to the standard TEASAR algorithm, we conducted a systematic ablation study. Starting from the kimimaro TEASAR, we incrementally add each proposed component in separate experiments. We observe a consistent improvement in performance with every addition in Table 5, ultimately achieving the lowest false merges and false splits values with our full model configuration.

## C.6. Training settings and hyperparameter analysis

**Training:** We conduct a series of experiments to evaluate the effects of different training settings. Quantitative results for the various datasets are presented in Table 6 and Table 7.

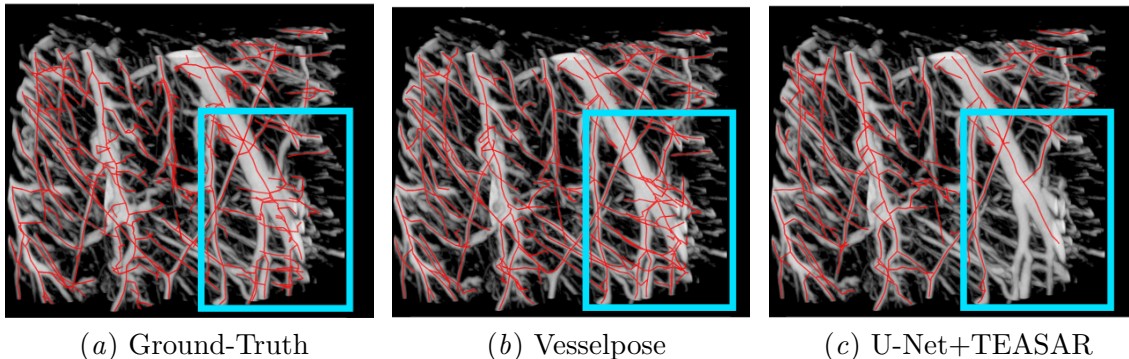

(a) Ground-Truth            (b) Vesselpose            (c) U-Net+TEASAR

Figure 9: **Qualitative results of the micro-CT data.** Illustrated is one 3D annotated crop from the raw micro-CT test data together with varying vessel skeleton graphs in red: (a) shows the annotated ground-truth skeleton; (b) shows the results of our proposed method; (c) shows the result of the baseline, which consists of a U-Net for foreground segmentation followed by the original TEASAR algorithm. Overall, our method accurately captures the vessel structures and aligns well with the ground-truth. In contrast, the baseline method fails to trace many vessel branches, in particular in the highlighted region within the blue box.

Table 5: Ablation study illustrating the contribution of individual components of our method and how they incrementally improve performance over a U-Net with standard TEASAR (Ronneberger et al., 2015; Sato et al., 2000). We add the following components step by step: support for multiple roots per connected component (multi-root); an additional penalty for tracing along vectors with small magnitudes (vec mag); an additional penalty for tracing in the same direction as the direction vector (vec dir); and adaptive masking to mark processed regions (adapt. mask). Results are shown for Vesselpose on the validation set of the Multi-Tree Synthetic dataset.

| Experiments | multi root | vec mag | vec dir | adapt. mask | Edges | | | FM | FS |
|---|---|---|---|---|---|---|---|---|---|
| | | | | | F1↑ | Prec↑ | Rec↑ | Abs.↓ | Abs.↓ |
| UNet+TEASAR | ✗ | ✗ | ✗ | ✗ | 0.46 | 0.63 | 0.36 | 52.4 | 54.1 |
| Ours | ✓ | ✗ | ✗ | ✗ | 0.71 | 0.78 | 0.64 | 38.3 | 38.0 |
| | ✓ | ✓ | ✗ | ✗ | 0.70 | 0.78 | 0.64 | 37.3 | 37.6 |
| | ✓ | ✓ | ✓ | ✗ | 0.75 | 0.79 | 0.73 | 33.6 | 35.0 |
| | ✓ | ✓ | ✓ | ✓ | 0.80 | 0.81 | 0.79 | 29.7 | 29.3 |

**TEASAR:** Similarly, we evaluate the TEASAR parameters—the penalty scale (1,000,000) and penalty exponent (16) (cf. Equation (2))—with results reported in Table 8 and Table 9. Our experiments indicate that varying penalty scale does not lead to substantial changes in TEASAR performance. We therefore retain the value used in the Kimimaro

Table 6: Quantitative comparison evaluating the effect of fixed tiles (in a sliding-window fashion) versus random crops from training samples in the multi-tree synthetic dataset.

| parameters | Edges | | | FM | | FS | |
|---|---|---|---|---|---|---|---|
| | F1↑ | Prec↑ | Rec↑ | Rel.↓ | Abs.↓ | Rel.↓ | Abs.↓ |
| sliding window | 0.79 | 0.80 | 0.78 | 0.008 | 33.36 | 0.008 | 36 |
| random crops (ours) | 0.80 | 0.81 | 0.79 | 0.007 | 30.33 | 0.007 | 29.3 |

Table 7: Quantitative comparison evaluating the effect of different training settings—data augmentation, learning rate, and foreground weighting—on single-tree datasets, using Trexplorer-Super metrics (Naeem et al., 2025), consistent with the corresponding baseline studies in Table 1.

| parameters | Dataset | Point Level | Branch Level | Graph Level | |
|---|---|---|---|---|---|
| | | F1↑ | F1↑ | Betti-0↓ | Betti-1↓ |
| no augmentation | Synthetic | 88.56 | 80.98 | 0 | 0 |
| 50% intensity shift (ours) | Synthetic | 92.28 | 81.28 | 0 | 0 |
| learning rate 0.001 | Parse2022 | 36.13 | 23.10 | 1.55 | 0 |
| learning rate 0.0001 (ours) | Parse2022 | 49.42 | 28.87 | 2.70 | 0 |
| w/o foreground weight | Parse2022 | 49.42 | 28.87 | 2.70 | 0 |
| w/ foreground weight (ours) | Parse2022 | 57.89 | 36.75 | 1.20 | 0 |

implementation of TEASAR (Silversmith et al., 2021). This choice is also consistent with the original TEASAR paper (Sato et al., 2000), where this parameter is described as being selected heuristically based on the skeleton segment. In contrast, penalty exponent has a more pronounced effect on the results. As the exponent increases, the edge-wise F1 score generally improves. However, for very large values, TEASAR begins to merge distinct trees, which is reflected in an increased Betti-1 error. Based on this trade-off, we selected the same value as used in both Kimimaro and the original TEASAR method.

## C.7. Sensitivity to vector prediction quality

We evaluate robustness to directional noise by perturbing the predicted vector field with an additive error term such that the error magnitude is proportional to the local predicted vector norm. Specifically, each original predicted direction vector $v$ is perturbed to $v' = v + \varepsilon \|v\| \cdot u$, where $\varepsilon \geq 0$ controls the noise level (noise-to-signal ratio) and $u$ is a random unit vector. We sweep $\varepsilon$ from 0 to 2.0 in steps of 0.1 and quantify performance using the edge-wise $F1$ score. As shown in Figure 10(a), our method remains stable over a broad range of perturbation strengths: Edge-wise F1 remains nearly constant for small to moderate noise

Table 8: Quantitative comparison of our modified TEASAR with varying penalty scale term (cf. Equation (2)). Results are shown for Vesselpose on the validation data of the Multi-Tree Synthetic dataset. We see that results are constant across different penalty scales.

| penalty scale | Edges | | | FM | | FS | |
|---|---|---|---|---|---|---|---|
| | F1↑ | Prec↑ | Rec↑ | Rel.↓ | Abs.↓ | Rel.↓ | Abs.↓ |
| $5 \times 10^3$ | 0.81 | 0.82 | 0.79 | 0.007 | 28 | 0.006 | 27 |
| $5 \times 10^4$ | 0.81 | 0.83 | 0.80 | 0.007 | 29 | 0.007 | 29 |
| $5 \times 10^5$ | 0.80 | 0.82 | 0.79 | 0.007 | 28 | 0.006 | 27 |
| $5 \times 10^6$ | 0.81 | 0.82 | 0.79 | 0.007 | 28 | 0.006 | 27 |
| $1 \times 10^6$ (ours) | 0.81 | 0.82 | 0.79 | 0.007 | 28 | 0.006 | 27 |

Table 9: Quantitative comparison of our modified TEASAR with varying penalty exponent (cf. Equation (2)) on VesselPose validation data from the Multi-Tree Synthetic dataset. Edge-wise F1 score, false merges, and false splits improve with increasing exponent; however, at very high values, TEASAR merges distinct trees, as reflected in the Betti-0 value.

| penalty exp. | Edges | FM | | FS | | Betti | |
|---|---|---|---|---|---|---|---|
| | F1↑ | Rel.↓ | Abs.↓ | Rel.↓ | Abs.↓ | Betti-0↓ | Betti-1↓ |
| 2 | 0.64 | 0.01 | 40 | 0.009 | 39 | 1 | 0 |
| 4 | 0.70 | 0.008 | 33 | 0.007 | 32 | 1 | 0 |
| 8 | 0.77 | 0.008 | 34 | 0.007 | 32 | 1 | 0 |
| 16 (ours) | 0.81 | 0.007 | 28 | 0.007 | 27 | 1 | 0 |
| 32 | 0.81 | 0.005 | 22 | 0.004 | 18 | 4 | 0 |

levels and degrades only gradually as $\varepsilon$ increases. A pronounced drop is observed only at very large noise ($\varepsilon > 1.0$), where the direction field becomes strongly corrupted as seen in Figure 10(d) and the reconstruction quality deteriorates more noticeably. Importantly, even for a higher noise level of our approach consistently outperforms the baseline TEASAR ($F1_{\mathrm{edge}} = 0.46$), indicating higher tolerance to directional uncertainty.

## C.8. Sensitivity to test time gaussian noise

We further assess robustness to test-time perturbations by adding voxel-wise Gaussian noise to the normalized raw image. Specifically, the noisy input is generated as

$$I' = \mathrm{clip}(I_{\mathrm{norm}} + \epsilon, 0, 1), \qquad \epsilon \sim \mathcal{N}(0, \sigma^2)$$

Table 10: Test time sensitivity study of Vesselpose under varying gaussian noise. Performance remains stable at low $\sigma$, but higher noise levels lead to small disconnected components, increasing false splits and Betti-1 error.

| sigma | Edges | FM | | FS | | Betti | |
|---|---|---|---|---|---|---|---|
| | F1$\uparrow$ | Rel.$\downarrow$ | Abs.$\downarrow$ | Rel.$\downarrow$ | Abs.$\downarrow$ | Betti-0$\downarrow$ | Betti-1$\downarrow$ |
| 0 | 0.81 | 0.007 | 28 | 0.006 | 27 | 1 | 0 |
| 0.03 | 0.81 | 0.007 | 31 | 0.007 | 30 | 1 | 0 |
| 0.06 | 0.81 | 0.008 | 32 | 0.007 | 31 | 1 | 0 |
| 0.09 | 0.80 | 0.008 | 34 | 0.008 | 23 | 1 | 0 |
| 0.12 | 0.80 | 0.01 | 43 | 0.01 | 45 | 2 | 0 |
| 0.15 | 0.78 | 0.009 | 38 | 0.02 | 127 | 94 | 0 |

where

$$I_{\mathrm{norm}} = \frac{I - I_{\min}}{I_{\max} - I_{\min}}$$

Here, $\sigma$ denotes the standard deviation of the Gaussian noise and controls the noise level. The effect of increasing noise levels during inference is reported in Table 10. For small values of $\sigma$, the performance remains largely stable, with no substantial degradation. However, at higher noise levels, we observe the emergence of several small disconnected components, which is reflected in the increase in false splits and Betti-1 error.

## C.9. Comparison with nnU-Net

We evaluate a variant of Vesselpose in which the U-Net backbone is replaced by nnU-Net (Isensee et al., 2021), a self-configuring segmentation framework that automatically determines architecture and training hyperparameters from dataset properties, requiring no manual tuning. We extend the framework to predict voxel-wise direction vectors as additional output channels alongside the foreground mask. The remaining pipeline, including the modified TEASAR algorithm and post-processing, is identical to the main method. Results on the Multi-Tree Synthetic dataset are reported in Table 11.

Table 11: Quantitative comparison of Vesselpose with a U-Net versus nnU-Net backbone on the Multi-Tree Synthetic dataset. We observe that nnU-Net performs slightly better on the edge metrics, but shows slightly worse false merge and false split errors than U-Net.

| Model | Edges | | | FM | | FS | |
|---|---|---|---|---|---|---|---|
| | F1↑ | Prec↑ | Rec↑ | Rel.↓ | Abs.↓ | Rel.↓ | Abs.↓ |
| Ours(U-Net) | 0.80 | 0.80 | 0.79 | **0.007** | **29.7** | **0.007** | **28.3** |
| Ours(nnU-Net) | **0.81** | **0.81** | **0.81** | 0.008 | 33.6 | 0.008 | 32.8 |

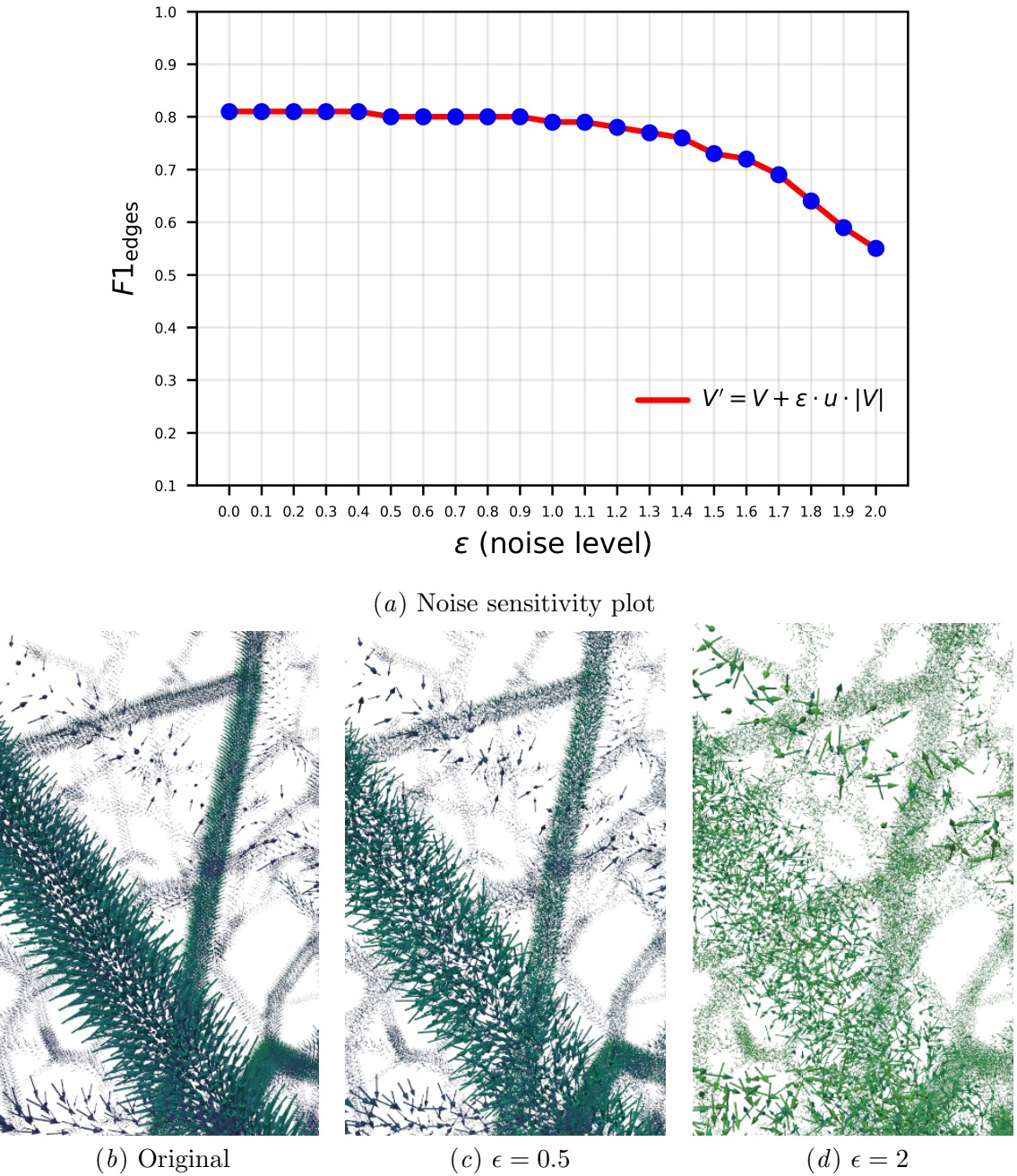

(a) Noise sensitivity plot

(b) Original          (c) $\epsilon = 0.5$          (d) $\epsilon = 2$

Figure 10: **Sensitivity to vector noise.** (a) The proposed method shows strong robust-
ness to vector noise: small to moderate perturbations (noise level $\varepsilon \leq 1.0$) of
the predicted vectors have no noticeable impact on the resulting edge-wise F1
score. (b–d) Visualization of the direction vector field under increasing noise
levels $\varepsilon \in \{0, 0.5, 2\}$. For clarity, all vectors are normalized. Dark blue indicates
low vector magnitude, while green indicates high vector magnitude.

