# OpenReview forum: "Vesselpose: Vessel Graph Reconstruction from Learned Voxel-wise Direction Vectors in 3D Vascular Images"
_MIDL.io/2026/Conference — MIDL 2026 Poster_

### Official Review · Reviewer_Tnmf · 2025-12-20

**Confidence:** 4
**Preliminary Rating:** 4
**Final Rating:** 5

**Summary:**

VesselPose proposes a novel two-stage approach for extracting topologically accurate vascular graphs from image data.
The method is based on a U-Net architecture that predicts both a binary vessel segmentation mask and voxel-wise direction vectors. These two outputs are subsequently combined in an adapted version of the TEASAR skeletonization algorithm to reconstruct a vascular graph. This formulation enables robust graph extraction even in the presence of closely spaced and parallel vessels. In addition to the proposed method, the authors introduce two new evaluation metrics and assess their approach across multiple experiments using both single-tree and multi-tree datasets.

**Strengths:**

The paper VesselPose is motivated by a clear, well-articulated, and clinically relevant objective, namely the extraction of topologically correct vascular graphs from image-based modalities. This motivation is well aligned with important downstream tasks in medical image analysis. Beyond this overarching motivation, the following strengths can be identified:

1.	The adapted penalty term is a technically well-designed contribution. By jointly accounting for vector magnitude and angular deviation, it leads to a noticeable improvement in vessel skeletonization quality, as evidenced by the reported results when compared to previous state-of-the-art methods.
2.	Extensive experimental validation is conducted across multiple publicly available datasets as well as an additional internal dataset. This breadth of evaluation increases confidence in the robustness and generalizability of the proposed approach.
3.	The related work section is comprehensive and clearly structured, providing a well-balanced overview of existing methods. Prior limitations are clearly highlighted, allowing the research gap addressed by the paper to be easily identified.
4.	The detailed appendix offers additional insights into the methodology, evaluation protocols, and experimental setups. This supplementary material demonstrates the depth of the authors’ work and enables a more thorough understanding of the proposed approach.
5.	Figures and visualizations are well designed and appropriately used. They are clearly structured and effectively support the explanation of the method and the interpretation of the results, thereby improving overall readability.
6.	The authors state that code, trained models, and related resources will be released upon publication, which is a strong positive aspect and significantly enhances the reproducibility and potential impact of the work.

**Weaknesses:**

The paper presents a solid contribution; however, several weaknesses limit the clarity and strength of the conclusions and could be addressed to further improve the work:

1.	Several methodological choices are insufficiently justified. For instance, the authors rely on a classical 3D U-Net architecture (originally introduced in 2015) without providing a clear rationale for not considering more recent and widely adopted alternatives, such as nnU-Net [1], or other foundation-model-based approaches that have shown strong performance in medical imaging (e.g., MedSAM [2] or SAM-Med3D [3]). Similarly, the choice of the scaling factor 1,000,000 in Equation (2) is not well motivated, particularly given that the original TEASAR formulation uses a different factor [4]. The rationale behind training with a batch size of 1 is also not discussed, although this choice can have a significant impact on optimization stability and generalization.
2.	The expressiveness and reliability of the reported results are limited by several factors. (i) The baseline results are not reproduced by the authors but instead taken from prior work, with the explanation that reproduction was not feasible. This makes direct comparisons harder to interpret. (ii) Some ground truth labels are automatically generated using the TEASAR algorithm itself, yet the potential bias or error propagation introduced by this procedure is neither analyzed nor discussed. (iii) Certain experiments rely on very small datasets (e.g., one data point for fine-tuning and three for validation and testing), which strongly limits statistical significance and the robustness of the conclusions.
3.	The overall methodological scope is somewhat limited. Large parts of the pipeline are built upon existing components, namely a standard U-Net architecture (with the extension of predicting direction vectors) and the TEASAR algorithm. As a result, the core novelty is concentrated mainly in the directional vector formulation, the adapted penalty term, and two relatively simple evaluation metrics. While these contributions are meaningful, the paper could benefit from a clearer discussion of how they advance beyond incremental improvements.
4.	No systematic hyperparameter optimization appears to have been conducted for the U-Net architecture or the training setup. Given the sensitivity of deep learning models to such choices, the lack of ablation studies or tuning experiments makes it difficult to assess whether the reported performance reflects the full potential of the proposed method.
5.	It is unclear, how disconnected fragments are evaluated for merging, as mentioned in Figure 3.

[1] Isensee, F., Jaeger, P. F., Kohl, S. A., Petersen, J., & Maier-Hein, K. H. (2021). nnU-Net: a self-configuring method for deep learning-based biomedical image segmentation. Nature methods, 18(2), 203-211.

[2] Ma, J., He, Y., Li, F., Han, L., You, C., & Wang, B. (2024). Segment anything in medical images. Nature Communications, 15(1), 654.

[3] Wang, H., Guo, S., Ye, J., Deng, Z., Cheng, J., Li, T., ... & Qiao, Y. (2024, September). Sam-med3d: towards general-purpose segmentation models for volumetric medical images. In European Conference on Computer Vision (pp. 51-67). Cham: Springer Nature Switzerland.

[4] Sato, M., Bitter, I., Bender, M. A., Kaufman, A. E., & Nakajima, M. (2000, October). TEASAR: tree-structure extraction algorithm for accurate and robust skeletons. In Proceedings the Eighth Pacific Conference on Computer Graphics and Applications (pp. 281-449). IEEE.

**Detailed Comments:**

1.	The statement that “obtaining accurate and complete reconstructions remains challenging” would benefit from additional justification or references to prior work to improve clarity and contextualize the challenges for the reader.
2.	All abbreviations should be introduced at first mention and then used consistently throughout the manuscript. For example, CT, MRI, ILP, DETR, DL, TP, FN, FP are not defined when first used, and ground truth is introduced only in Section 6 despite earlier usage. A thorough check of all abbreviations is recommended.
3.	Since the rat CT dataset is not publicly available, it would improve reproducibility to report the acquisition parameters, including scanner type, spatial resolution, kernel, kVp, and tube current.
4.	The training hardware should be specified to allow readers to better reproduce the experiments and understand potential computational limitations.
5.	Section 3.2, the first sentence should include an explicit reference rather than referring the reader solely to the Appendix, which would improve clarity and readability.
6.	Minor typographical issues were observed, such as capitalizing “However” after a dash on page 2. Lowercase is more appropriate in this context as the sentence is not terminated by the dash.
7.	For better readability, it is recommended to include non-breaking spaces before parentheses to prevent brackets from appearing at the beginning of a line. This improves the visual presentation of equations, citations, and in-text references.

**Justification Of Final Rating:**

Given the thorough rebuttal, the acknowledgment of limitations, and the planned additions, I raise my rating from 4 (weak accept) to 5 (strong accept). I emphasize that the authors should ensure all promised changes are fully incorporated in the camera-ready version to strengthen the manuscript.

**Justification Of The Preliminary Rating:**

My evaluation is primarily based on the strong experimental results demonstrated across multiple metrics and diverse datasets. The methodology is thoroughly validated, showing that the proposed approach outperforms existing methods in this application domain. The paper is well-structured, clearly written, and easy to follow, which facilitates understanding and reproducibility.

While some methodological choices are not fully justified in detail and the architecture is relatively simple, I do not consider these issues sufficient to detract from the overall contribution. The functionality and effectiveness of the method outweigh the lack of architectural complexity, and the work provides a meaningful advancement for vascular graph extraction from imaging modalities.

**Questions To Address In The Rebuttal:**

In addition to the points raised regarding the limitations of the work, I would appreciate if the authors could comment on the following aspects:

1.	Handling of complex graph structures: How does the network deal with challenging regions such as bifurcations in the vascular graph? Since only a single direction vector is predicted per voxel, could directional ambiguity at these points lead to errors, and how is this addressed?

2.	Robustness to noisy input data: How sensitive is the U-Net-based pipeline to noise in the input images? Does the method maintain accurate predictions and skeletonization under such conditions, and are there strategies to improve robustness if performance degrades?

---

> ### Author Response · Authors · 2026-01-24
>
> Thank you very much for your thorough review and detailed comments. We will incorporate all “Detailed Comments” in the camera-ready version of the paper. Regarding your concerns about the comparability of our results with modern ViT-based graph methods and robustness to noisy input data, we refer to our response to reviewer XRc3, who raised similar points. In addition, to address your concern about unclear handling of unconnected components, we have moved the corresponding paragraph on false split postprocessing from the supplemental material to the main text in the revised version of the paper (Section 2.2). Below, we address your remaining main concerns.
>
> • **Justification of methodological choices.**
>
> We will add nnUnet in the camera-ready version of the paper. However so far our existing pipeline is on par with preliminary results based on nnUnet. We will also include additional clarifying details regarding the TEASAR scaling factor and the used batch size in the camera-ready version.
>
> • **Expressiveness and reliability of the reported results.**
>
> In the revised main text, we have explicitly stated the potential bias that may favor our method on the PARSE dataset, as well as the limited statistical significance of the results on the small multi-tree micro-CT dataset (Sections 4.2 and 4.3, respectively, highlighted in red). We thank the reviewer for pointing out these important considerations.
>
> • **Limited methodological contribution.**
>
> Our main aim is to advance vessel centerline reconstruction, a task for which biologists still often rely on manual or semi-automated approaches in practice [1,2]. To this end, we transfer a highly successful paradigm—leveraging pixelwise topological information—from cell segmentation and tracking to vessel centerline reconstruction, requiring substantial remodeling and several methodological advances. The resulting approach is robust, reproducible, and easy to adapt, and it demonstrates strong performance across multiple datasets. Beyond the reconstruction method itself, we view the introduction of hierarchical matching and topology-aware evaluation metrics as an important contribution toward more meaningful and fair benchmarking. Finally, to further promote reproducibility, we will release trained models and qualitative results, enabling the community to build directly upon our work.
>
> • **Missing systematic hyperparameter optimization.**
>
> We did perform hyperparameter tuning on the validation set during pipeline development, including U-Net architecture choices (depth and downsampling), TEASAR scale and constant ranges, angular deviation thresholds, as well as training hyperparameters such as learning rate and foreground voxel weighting for PARSE. In the camera-ready version supplement, we will include a systematic validation study covering these parameters (and additional ones if necessary).
>
> • **Handling of complex graph structures.**
>
> Potential directional ambiguities at bifurcations are resolved by defining a consistent flow direction from endpoints toward the root, inspired from the vector definition in Malin-Mayor et al. [3]. With this convention, a single direction vector per pixel is well-defined to represent the local topology also in the case of bifurcations. We added two sentences in the introduction of the revised paper to state this clearly (highlighted in red).
> Nevertheless, we have identified two failure cases: (i) the erroneous merging of two spatially close roots, and (ii) missing small branches near the leaf nodes, where vessel diameters are typically very thin. To transparently illustrate these limitations, we have moved the corresponding qualitative figure from the multi-tree synthetic dataset to the main paper.
>
>
> [1] Pamela E. Rios Coronado et al., CXCL12 drives natural variation in coronary artery anatomy across diverse populations, Cell, 2025
>
> [2] Mireia Pampols-Perez et al., Mechanosensitive piezo2 channels shape coronary artery development. Nature Cardiovascular Research, 2025
>
> [3] Caroline Malin-Mayor et al., Automated reconstruction of whole-embryo cell lineages by learning from sparse annotations, Nature Biotechnology, 2023

---

> > ### Comment · Reviewer_Tnmf · 2026-01-29
> >
> > I thank the authors for their detailed responses and clarifications. The rebuttal addresses most of my concerns and demonstrates a strong commitment to improving the manuscript. However, several important aspects are currently only promised for the camera-ready version. For example, no concrete results of the reported hyperparameter optimization are provided, which leaves the sensitivity of the method to hyperparameter choices unclear at this stage. I therefore strongly encourage the authors to ensure that all promised analyses and clarifications are fully included in the camera-ready version to further strengthen the paper.

---

### Official Review · Reviewer_kB1n · 2026-01-05

**Confidence:** 4
**Preliminary Rating:** 4
**Final Rating:** 5

**Summary:**

The paper addresses vessel graph extraction from vascular imaging, with a focus on preserving topological correctness, particularly by reducing false merges in vascular graphs through vector direction prediction. In addition, it introduces a novel topology-aware metric to quantitatively evaluate the extracted graphs.

**Strengths:**

the problem is clearly motivated and well explained. The authors provide a clear analysis of the limitations of existing state-of-the-art methods, particularly regarding false merges, and convincingly describe how insights from a related domain inspired their proposed solution.

**Weaknesses:**

To me the main weakness of the paper is that several important methodological details are only described in the supplementary material, making the approach difficult to fully understand when reading the main text alone.
In particular, the supplementary material mentions a vascular-specific metric that appears closely aligned with the authors’ stated goals; however, this metric is neither discussed in the main paper nor used or compared in the evaluation.

**Detailed Comments:**

The caption of Figure 4 refers to the ground truth graph in yellow; however, no yellow elements are clearly visible in the figure, making this description confusing.

**Justification Of Final Rating:**

The problem is well motivated, the main idea is interesting, and the proposed metric is a valuable contribution. The authors have successfully addressed the concerns raised in my review during the rebuttal, in particular by clarifying methodological details and reducing reliance on the supplementary material.

**Justification Of The Preliminary Rating:**

The problem is well motivated, the main idea is interesting, and the proposed metric is a valuable contribution. The paper would benefit from clearer descriptions of some methodological details and reduced reliance on the supplementary material.

**Questions To Address In The Rebuttal:**

For me mainly it's a matter of reorganizing some things in the main body/supplemental, and including in the main body(or explaining why not) the metric mentioned before

---

> ### Author Response · Authors · 2026-01-24
>
> Thank you very much for your helpful review.
> To improve readability, we have moved the false split postprocessing paragraph as well as the mathematical definitions for edge-wise F1, precision and recall from the supplementary material into the main text (Sections 2.2 and 3.2 respectively).
> When referring to a “vascular-specific metric,” could you please clarify whether this corresponds to false merges and false splits? If so, we have added Table 2, which also evaluates the Single-Tree datasets using our method and metrics. For the Multi-Tree datasets, this evaluation was already included in Table 3.

---

### Official Review · Reviewer_XRc3 · 2026-01-13

**Confidence:** 4
**Preliminary Rating:** 5

**Summary:**

This paper proposes a l framework for reconstructing topologically accurate 3D vascular graphs by predicting voxel-wise direction vectors jointly with vessel segmentation, and assembling them into centerline graphs via a direction-guided extension of the TEASAR algorithm.

Departing from the classical segment-then-skeletonize paradigm, the method treats topology as a first-class modeling objective rather than a downstream repair problem.

The authors further introduce a hierarchical graph matching strategy and define false splits and false merges as interpretable, topology-aware evaluation metrics. The approach is validated across synthetic and real datasets, including both single-tree and multi-tree vascular scenarios, demonstrating consistent improvements over segmentation-based pipelines and recent image-to-graph methods

**Strengths:**

I like this paper very much. The method can be interpreted as learning an implicit Morse-Smale structure over the vessel volume, replacing handcrafted distance-transform–based skeletonization with a learned flow field.

Specific strengths:

x. Conceptual clarity and correctness. The paper identifies a fundamental mismatch in existing pipelines: segmentation masks do not encode connectivity. Predicting direction fields that explicitly encode centerline orientation and rootward flow is a principled and well-justified solution.

x Good reuse of classical algorithms. Rather than discarding TEASAR, the authors augment it with learned geometric priors (vector magnitude and angular consistency). This hybrid learning–algorithmic approach is computationally efficient, interpretable, and scalable.

x. Strong handling of multi-tree scenarios. Most recent image-to-graph methods implicitly assume single-tree structures. Vesselpose explicitly supports multiple roots, multiple trees, and false-merge recovery, making it substantially more realistic for real-world vascular data.

x. Meaningful evaluation metrics. The introduction of false splits and false merges is a major contribution. These metrics capture structural damage that edge-wise F1 alone cannot, and the hierarchical matching strategy avoids the brittleness of naive nearest-neighbor assignments.

x. Solid empirical validation. The method is evaluated on diverse datasets (synthetic, CT, micro-CT) and consistently improves topological correctness. The qualitative examples convincingly demonstrate reduced false merges in closely apposed vessels.

**Weaknesses:**

I only identify two weaknesses:

x. Limited comparison to modern end-to-end graph methods.
Comparisons to recent transformer-based approaches (e.g., Trexplorer variants) are constrained by evaluation protocol differences. A deeper analysis of failure modes where learning-based tracking fails versus where Vesselpose fails, would strengthen the positioning.

x. Sensitivity to vector prediction quality.
The method’s success hinges on accurate direction vectors. However, there is limited discussion on how errors in vector magnitude or angular prediction propagate into the final graph, especially in low-contrast or discontinuous vessel regions.

x. (very minor)
An interesting future direction would be to explore whether parts of the vector-guided reconstruction could be relaxed into a differentiable formulation (e.g., via implicit potentials or soft path costs), though it is not obvious that this would preserve the current method’s stability.

**Detailed Comments:**

Please see the above comments.

**Justification Of The Preliminary Rating:**

The contribution of this work (methodology novelty in topology + empirical validation on mutiple and diverse datasets) is more significant than other papers in my batch.
Hence, I recommend strong accept.

**Questions To Address In The Rebuttal:**

Please see the above comments.

---

> ### Author Response · Authors · 2026-01-24
>
> Thank you very much for your positive review. Below, we address your concerns in detail.
>
> • **Limited comparison to modern end-to-end graph methods.**
>
> To compare with the Trexplorer-Super method, we ran their evaluation protocol and code, and the reported results are therefore directly comparable (Table 1). We agree that a qualitative comparison would be desirable. However, despite substantial effort, we were unable to reproduce the results as published for several recent end-to-end graph methods due to missing information in the publicly available repositories. Our reproduced scores differed significantly. Instead, to facilitate future qualitative comparisons, we will make our trained model and the corresponding predicted results publicly available with the camera-ready version of the paper. Moreover, we have moved the qualitative results figure for the synthetic multi-tree dataset (Figure 5) to the main paper that also shows failure cases of our method.
>
> • **Sensitivity to vector prediction quality.**
> We have added an analysis in section C.6 of the supplementary material in which we introduce increasing levels of noise to the predicted direction vectors and evaluate the edge-wise F1 score. We find that our method shows strong robustness to vector noise: small to moderate perturbations of the predicted vectors have no noticeable impact on the F1 score. In particular, even under noisy perturbations of the direction vector field our method consistently outperforms the baseline TEASAR. For the camera-ready version, we plan to add a similar analysis by introducing noise to images at test time and assessing the resulting robustness.
>
> • **Relaxation of the vector-guided reconstruction into a differentiable formulation.**
> We agree that relaxing parts of the vector-guided reconstruction into a differentiable formulation is an interesting direction for future work. In principle, this could enable fully end-to-end training by allowing backpropagation through a TEASAR-like procedure. However, developing such a formulation while preserving the stability of the current method is beyond the scope of this paper and would likely require a dedicated theoretical investigation.

---

> > ### Comment · Reviewer_kB1n · 2026-01-30
> >
> > Thank you for the detailed response and for moving key methodological details from the supplementary material into the main text; this indeed improves the readability of the paper.
> > Additionally, I would like to point out a minor but still unresolved issue in Figure 6 (formerly Figure 4): the caption refers to the ground truth graph being shown in yellow, but no yellow elements are clearly visible in the figure. Clarifying or correcting this description would avoid confusion for the reader.
> > Overall, the revisions are appreciated and move the paper in a positive direction.

---

> > > ### Author Response · Authors · 2026-02-01
> > >
> > > Thank you very much for pointing this out again. Indeed, this is a mistake, and we will change “yellow” to “dark red” in the description of Figure 6 in the camera-ready version of the paper.

---

> > ### Comment · Reviewer_XRc3 · 2026-01-31
> >
> > Thank you for the response. All issues are resolved.

---

### Author Rebuttal · Authors · 2026-01-24

**Rebuttal:**

Please find the revised manuscript here, with changes highlighted in red.

**Supporting Material:**

/attachment/2274044030fbe53cb0fc393f4927a33970f9d5ad.pdf

---

### Meta-Review · Area_Chair_GHUA · 2026-02-09

**Recommendation:** Accept (Oral)
**Confidence:** 5

**Metareview:**

All reviewers recommend a strong accept, praising the framework’s innovative use of learned direction fields to ensure topological accuracy and its robust validation across diverse vascular datasets.

---

### Decision · Program_Chairs · 2026-02-13

Accept (Poster)